How word-beginnings constrain the pronunciations of word-ends in the reading aloud of English: the phenomena of head- and onset-conditioning

Ulicheva Anastasia 1 ulicheva@connect.hku.hk
Coltheart Max 2
1 Centre for Communication Disorders, The University of Hong Kong , Hong Kong, SAR China
2 ARC Centre of Excellence in Cognition and its Disorders (CCD), and Department of Cognitive Science, Macquarie University , Sydney , Australia
Barnhart Anthony
Electronic publication date: 2015 Dec 3
Publication date: 2015
Volume: 3
Electronic Location ID: e1482
Received 2015 Aug 30; Accepted 2015 Nov 17
Copyright: © 2015 Ulicheva and Coltheart
Copyright year: 2015
Copyright holder: Ulicheva and Coltheart
License: This is an open access article distributed under the terms of the Creative Commons Attribution License, which permits unrestricted use, distribution, reproduction and adaptation in any medium and for any purpose provided that it is properly attributed. For attribution, the original author(s), title, publication source (PeerJ) and either DOI or URL of the article must be cited.
License URL: https://creativecommons.org/licenses/by/4.0/

Keywords: Word naming, Spelling-to-sound correspondence, Regularity, Consistency, Phonology

Funding: Australian Research Council Centre of Excellence in Cognition and its Disorders CE110001021 This research was supported by the Australian Research Council Centre of Excellence in Cognition and its Disorders (project number CE110001021). The funders had no role in study design, data collection and analysis, decision to publish, or preparation of the manuscript.

==============================
Background. A word whose body is pronounced in different ways in different words is body-inconsistent. When we take the unit that precedes the vowel into account for the calculation of body-consistency, the proportion of English words that are body-inconsistent is considerably reduced at the level of corpus analysis, prompting the question of whether humans actually use such head/onset-conditioning when they read.

Methods. Four metrics for head/onset-constrained body-consistency were calculated: by the last grapheme of the head, by the last phoneme of the onset, by place and manner of articulation of the last phoneme of the onset, and by manner of articulation of the last phoneme of the onset. Since these were highly correlated, principal component analysis was performed on them.

Results. Two out of four resulting principal components explained significant variance in the reading-aloud reaction times, beyond regularity and body-consistency.

Discussion. Humans read head/onset-conditioned words faster than would be predicted based on their body-consistency and regularity only. We conclude that humans are sensitive to the dependency between word-beginnings and word-ends when they read aloud, and that this dependency is phonological in nature, rather than orthographic.

Introduction

The words BUSH and PUSH are classified as irregular words, because the grapheme1 U in these words is not given its most common phonemic pronunciation (which is as in RUST or BUNK). Words which contain the most common phonemic assignment for each grapheme are regular words. The judgment about the regularity status of a grapheme is performed based on a predefined set of grapheme-phoneme correspondence rules, such as that implemented in the Dual-Route Cascaded model (DRC; Coltheart et al., 2001).

The words BUSH and PUSH are also classified as inconsistent words, because the orthographic body—USH in these words has a different pronunciation in other words (e.g., RUSH, LUSH). For this reason—USH is classified as an inconsistent body.2 Words whose bodies have the same pronunciation in all words containing that body are consistent words.

In studies of reading aloud, irregular words produce longer latencies than regular words (Baron & Strawson, 1976; Gough & Cosky, 1977; Stanovich & Bauer, 1978), and inconsistent words produce longer latencies than consistent words (Glushko, 1979; Treiman et al., 1995; Jared, 2002; Chateau & Jared, 2003), and these effects are independent of each other (Jared, 2002). Therefore, human readers rely on units of small size (graphemes and phonemes), as well as on units of larger size (bodies and rimes) when they read aloud.

Are graphemes and phonemes, as well as bodies and rimes, the only units represented in the human reading system? To this end, Treiman et al. (1995) reported that antibody3-consistency did not influence reading times. The authors concluded that initial consonants are less closely linked to the vowel grapheme, and thus, do not represent a reading unit. The consistency of the head alone (without the vowel), on the contrary, was reported as influencing oral reading times (Treiman et al., 1995; Balota et al., 2004). However, since the regularity of grapheme-to-phoneme correspondence was not rigorously controlled in studies that found head-consistency effects, it is possible that the concept of the head as a reading unit is not required to explain them, contrary to what Treiman et al. suggested (also see Yap & Balota, 2009).

Venezky & Massaro (1987) suggested an elaboration of the concept of consistency which is potentially important for understanding reading aloud but which was never subsequently pursued. They noted (p. 167–168): “Bush and push both contain what most people would classify as irregular pronunciations of u (cf. flush, rush, crush, dull, hull). However, after a non-nasal bilabial, u corresponds regularly to /u/ in some following consonant environments (e.g., full, pull, bull, bushed, butcher, fulsome). Does this make the u in bush and push regular? Semi-regular?”

What these authors meant by the expression “following consonant environments” was orthographic bodies. So what they were suggesting here is that we should allow for the possibility that the assignment of a phonological rime to an orthographic body might be influenced by the phoneme immediately preceding the body, i.e., the last phoneme of the word’s onset. We will refer to this hypothetical phenomenon as “onset-conditioning” of body-to-rime relationships. An example of a body-to-rime rule incorporating onset-conditioning would be: “–USH is pronounced as in BUSH when it follows a non-nasal bilabial”. If this rule is used, BUSH and PUSH become consistent rather than inconsistent words.

Although Venezky & Massaro (1987) treated this phenomenon as phonological in nature, it could be orthographic. It is possible that the dependency exists between heads and bodies (the last grapheme of the head, B, conditions the body pronunciation: we will refer to this as “head-conditioning”). Alternatively, the dependency might be between onsets and rimes (the past phoneme of the onset, /b/, conditions the body pronunciation; we refer to this as “onset-conditioning”).

The aim of our paper is to investigate the neglected Venezky–Massaro conjecture that one factor influencing the mapping of phonology-to-orthography in reading aloud is that the last phoneme of the onset constrains the phonology of the rime. We pursue this aim by investigating three questions:

(1) Is head/onset-conditioning actually evident when one analyses English spelling-sound corpora?

(2) If it is, does head/onset-conditioning have psychological reality, i.e., are people’s responses influenced by it in the reading-aloud task?

(3) If they are, is it just head-conditioning, or just onset-conditioning, or both, that is important? That is, is the effect orthographic or phonological or both?

It is essential to distinguish the concept of head/onset-conditioning from the concepts of head-consistency and antibody-consistency. Head/onset-conditioning refers to the constraint exerted by a single element (the last grapheme of the head or the last phoneme of the onset) on the part that immediately follows it (the vowel or the body). In contrast, head-consistency and antibody-consistency refer to orthographic segments that can consist of multiple elements (the head of STRAIN consists of three graphemes STR–; its antibody consists of four graphemes STRAI–).

Furthermore, the concepts of head-consistency and antibody-consistency do not refer to constraints exerted by the head or the antibody on any other part of the letter-string; instead, these two concepts refer to other words having the same head or antibody. Hence, the concepts of head-consistency and antibody-consistency are similar to each other, and both are very different from the concept of head/onset-conditioning.

Materials & Methods

Measures

For monosyllabic English words (7,755 words in the vocabulary of the English DRC model; Coltheart et al., 2001) their type body-consistency was calculated using the traditional approach—that is, the number of words where the orthographic body is pronounced in the same way as in the target word was divided by the total number of words where this body occurs. For example, type body-consistency for the word BUSH is low (0.19). There were 1,744 different bodies in the corpus. We were interested to see whether inconsistency in mapping bodies to rimes is reduced by taking into account the phoneme or grapheme that precedes the vowel (that is Question (1) above). This was investigated using four additional measures of body-consistency.

To establish whether the conditioning is orthographic or phonological in nature, the critical element preceding the vowel was either orthographic (the preceding grapheme; measure 1) or phonological (the preceding phoneme or its phonological properties; measures 2a–c). For onset-conditioning, either individual phonemes were used for calculation (measure 2a), or else phonemes grouped by phonological features (by place and manner of articulation, in measure 2b; by manner of articulation, in measure 2c). The assumption for creating measures 2b and 2c was that phonemes that share phonological features may trigger the same type of onset-conditioning.

In this work, we focused on place and manner of articulation, although other phonological features, such as nasality, might be important as well (Venezky & Massaro, 1987). All calculations were based on type rather than token measures. In this work we pursed the question of psychological reality of onset-conditioning and the methodology that we adopted would not allow for disentangling type and token effects. Thus, we focus only on type measures of onset-conditioning.

Measure 1 was body-consistency conditioned by the last grapheme in the head. It was calculated in the following way. All 1,744 subsets of words with the same orthographic body were investigated (for example, for the word SWEAT with type body-consistency of 0.12, this subset consisted of 17 words and included body-enemies4 like SEAT as well as body-friends like THREAT). Words in each subset were classified into groups depending on the last grapheme in the head. There are 36 different graphemes that can appear before the vowel grapheme in English monosyllables, including single-letter graphemes such as S and multi-letter graphemes such as TH. So 36 groups, one for each grapheme, were analysed. For example, with the body –EAT, 13 such grapheme-based word groups were formed. Then the type measure of head-conditioned body-consistency was obtained by taking the sum of all words in this group where the body is pronounced in the same way as in the target word (i.e., head-conditioned friends) and dividing this number by the total number of words in this group (i.e., head-conditioned neighbours). Each word was counted as its own neighbour. In case of SWEAT, no other word in its body-neighbourhood has the last grapheme in the onset W. Thus, the value of measure 1 for SWEAT is 1, that is, this word is consistent when the last grapheme in the head is taken into account, though inconsistent otherwise. Note that taking the last phoneme in the onset is not beneficial in this case, because SWEAT would be grouped with WHEAT, and the bodies of these two words are pronounced differently (resulting in the phoneme-conditioned body-consistency value of 0.5).

Measure 2a was body-consistency conditioned by the last phoneme in the onset (the calculation was analogous to that of measure 1). There are 22 different phonemes that can appear before a vowel in English monosyllables. To illustrate, in the case of SNOW with type body-consistency of 0.57, three words in its body-neighbourhood have /n/ as the last phoneme in the onset: KNOW, NOW and SNOW itself. Thus, the value of measure 2 for SNOW is 0.67; that is, this word is more consistent when the last phoneme in the onset is taken into account than when no onset-conditioning (body-consistency) or head-conditioning (measure 1) is used. Note that knowing the last grapheme in the onset is not beneficial in this case, because the final grapheme in KNOW, KN, is not the same as in SNOW, N, and thus, KNOW and SNOW are not grapheme-conditioned friends by measure 1 (this results in a lower value for measure 1 for SNOW, i.e., 0.5).

Measure 2b was body-consistency conditioned by the place and manner of articulation of the last phoneme in the onset. The grouping by place and manner of articulation resulted in 15 classes of phonemes. Eight classes contained only one phoneme (e.g., the velar approximant /w/) and seven classes contained two phonemes each (e.g., bilabial stops, /b/ and /p/; labio-dental fricatives, /f/ and /v/). Note that this measure reflects phonological conditioning, because a similar grouping would be impossible in orthography. The calculation was analogous to that of measures 1 and 2a. For example, in the case of PEAR with body-consistency of 0.26, two other words in its body-neighbourhood have the last phoneme that belongs to the class “bilabial stops,” SPEAR and BEAR. By measure 2b, BEAR is an onset-conditioned friend of PEAR, while SPEAR is an onset-conditioned enemy. Thus, the value of measure 2b for PEAR is 0.67, that is, this word is more consistent when the place and the manner of articulation of the last phoneme in the onset is taken into account, and less inconsistent otherwise. Note that knowing the last grapheme or the last phoneme in the onset does not grant the same advantage in this case, because BEAR is not an onset-conditioned friend of PEAR by measures 1 and 2, and this would result in a lower value, i.e., 0.5.

Measure 2c was body-consistency conditioned by the manner of articulation of the last phoneme in the onset. The grouping by place and manner of articulation resulted in six classes of phonemes. One class contained only one phoneme (e.g., lateral /l/), other classes contained more than one phoneme each (e.g., three phonemes in the class of approximants, six phonemes in the class of stops). The calculation was analogous to that of measures 1–2b. For example, in case of BROW with the body-consistency of 0.43, seven other words in its body-neighbourhood have the last phoneme that belongs to the class “approximants,” e.g., ROW and WOW. By measure 2c, BROW is more consistent (0.5) than by any other measure, because it is only by this measure that WOW is an onset-conditioned friend of BROW.

Question 1: Is head/onset-conditioning commonly seen when one analyses English spelling-sound corpora?

In our corpus, the number of words with traditionally-calculated type body-consistency less than 1 is 2,302 (29.9%). In comparison, for the four measures 1–2c, there are 269 (3.5%), 299 (3.9%), 366 (4.7%), and 917 (11.8%) words in the corpus of monosyllabic words that are inconsistent, i.e., that have a head/onset-conditioned value for body-consistency that is less than 1. These results show that far fewer words are inconsistent when head/onset-conditioning is taken into account than when it is not. This indicates that head/onset-conditioning is relatively common in English.

We can obtain examples of onset-conditioning by selecting words that are body-consistent with respect to, say, measure 2c, but are body-inconsistent using traditionally-calculated type body-consistency. For example, the body –OOD after nasals (/m/, /n/) is pronounced /–ud/ (MOOD, SNOOD), but after stops (/g/, /t/) is pronounced (GOOD, STOOD). The body –OWN after fricatives (, /s/ in SHOWN, SOWN) and nasals (/m/, /n/ in KNOWN, MOWN) is , while after stops (/d/, /g/, /t/ in DOWN, GOWN, TOWN) it is .

In theory, then, readers would profit by exploiting head/onset-conditioning when reading aloud. But do they do so in practice: does this property of letter/phoneme-strings have psychological reality?

Question 2: Does head/onset-conditioning have psychological reality, i.e., are people influenced by it when reading aloud?

In order to answer this question, we analysed naming latencies to English words from the English Lexicon Project (ELP; Balota et al., 2007), a database that includes naming latencies for over 40,000 words collected from 443 participants. The question of interest was: do words that have higher body-consistency values when the last element in the onset is taken into account elicit shorter RT compared to those that do not (when other commonly studied psycholinguistic variables like written frequency and length are taken into account)?

The following psycholinguistic variables were extracted from ELP: standardised RT (I_NMG_Zscore), word length (Length), written-word frequency (log-transformed, Log_Freq_HAL), orthographic N-size (Ortho_N), phonological N-size (Levenstein measure, PLD), and positional bigram frequency (BG_Freq_By_Pos). Body-consistency was calculated in the traditional way (Type_Body; see ‘Introduction’). Token_Body measure was also calculated using word frequencies from the HAL corpus (available from the ELP database): summed frequencies of body-friends were divided by summed frequency of body-neighbours. Grapheme-to-phoneme regularity (Reg) was defined based on the GPC rule set. The values were extracted from the DRC model (Coltheart et al., 2001). This variable was categorical: if a word consisted of grapheme-to-phoneme correspondences, all of which were most common correspondences for these graphemes, then this word was regular, and it was irregular otherwise. In order to correct for potential voice key biases, we took the quality of the initial phoneme into account following the classification adopted in Balota et al. (2004). More specifically, we added 13 categorical variables that denoted the presence or absence of one of the following phonological features: affricative, alveolar, bilabial, dental, fricative, glottal, labiodental, liquid, nasal, palatal, stop, velar, and voiced. The number of letters in the onset was also included as a predictor (Treiman et al., 1995). Finally, type and token CV consistency were calculated. As in the case with token body-consistency, token frequencies were taken from the HAL corpus. Following Treiman et al., CV was taken to be all graphemes in the onset and the vowel grapheme (including split graphemes; for example, CV in WHOLE is WHO.E, and WHOLE is CV-inconsistent, because it has an enemy, WHOSE). Word length, type and token consistency, orthographic and phonological N-size values were centered (Baayen, 2008). Bigram frequency was logarithm-transformed.

Linear models of increasing complexity were implemented in R (R Development Core Team, 2011). Words that were not included in ELP or had missing RT values were discarded (31%). Duplicated entries were excluded (69 words). In order to ensure that our results are not due to some obvious cases of orthographic contextual dependency like WA (as in WANT), 164 words that contained patterns WA, GI, GE, CI, and CE were excluded. The resulting dataset contained 5,123 observations. The dependent variable in all models was the standardised RT.

Model A included word length, written word frequency, orthographic and phonological N-sizes, positional bigram frequency, type and token body-consistency, regularity, length of the onset in letters, CV type and token consistency, and variables coding the quality of the initial phoneme as predictors. The formula in R was lm(I_NMG_Zscore ∼ Length + Log_Freq_HAL + Type_Body + Token_Body + Ortho_N + PLD + BG_Freq_By_Pos + Reg + Onset_Length + Type_CV + Token_CV + Affricative + Alveolar + Bilabial + Dental + Fricative + Glottal + Labiodental + Liquid + Nasal + Palatal + Stop + Velar + Voiced).

Model B was identical to Model A, except that four measures of onset-conditioning were added as predictors: lm(I_NMG_Zscore ∼Length + Log_Freq_HAL + Type_Body + Token_Body + Ortho_N + PLD + BG_Freq_By_Pos + Reg + Onset_Length + Type_CV + Token_CV + Affricative + Alveolar + Bilabial + Dental + Fricative + Glottal + Labiodental + Liquid + Nasal + Palatal + Stop + Velar + Voiced + Measure_1 + Measure_2a + Measure_2b + Measure_2c).

Model C was obtained by excluding non-significant predictors from Model B. Linear nested models were compared using R’s anova() command.

Results of ELP analyses (original variables)

Model B provided a better fit to the data than Model A (F (4, 5,094) = 7.883, p < 0.0001). Among the variables of interest (measures 1–2c) in Model B, only measure 1 and measure 2a produced significant effects on human RT. Thus, Model C included these two measures and did not include measures 2b and 2c that were not significant in Model B; we also excluded non-significant Token_Body, BG_Freq_By_Pos, CV_type, and Liquid variables. Models B and C statistically did not differ from each other (p > 0.5), so the simpler Model C was chosen Table 1 presents the output of this final model.

Table 1 Output of the final model with original predictors.

	B	SE	t value	p value	
(Intercept)	0.086558	0.034528	2.507	p < 0.05	
Length	0.030646	0.00352	8.707	p < 0.001	
Log_Freq_HAL	−0.042277	0.001188	−35.597	p < 0.001	
Type_Body	−0.016267	0.003037	−5.356	p < 0.001	
Ortho_N	−0.026592	0.003532	−7.529	p < 0.001	
PLD	−0.02056	0.003284	−6.261	p < 0.001	
Reg	−0.071896	0.008376	−8.584	p < 0.001	
Bilabial	−0.25616	0.032962	−7.771	p < 0.001	
Dental	−0.404453	0.038481	−10.51	p < 0.001	
Labiodental	−0.40326	0.034641	−11.641	p < 0.001	
Alveolar	−0.196567	0.03216	−6.112	p < 0.001	
Palatal	−0.106499	0.015876	−6.708	p < 0.001	
Velar	−0.214151	0.032675	−6.554	p < 0.001	
Glottal	−0.540465	0.035475	−15.235	p < 0.001	
Nasal	0.119415	0.012195	9.792	p < 0.001	
Stop	0.093519	0.009228	10.134	p < 0.001	
Affricate	0.288411	0.022591	12.767	p < 0.001	
Fricative	0.318887	0.011771	27.091	p < 0.001	
Voiced	−0.040608	0.006822	−5.952	p < 0.001	
Onset_Length	0.009537	0.003371	2.829	p < 0.01	
CV_token	0.005074	0.002563	1.979	p < 0.05	
Measure_1	0.039912	0.007926	5.036	p < 0.001	
Measure_2a	−0.043066	0.008027	−5.365	p < 0.001	

The analysis of the ELP reading-aloud data showed that a model without the four onset-conditioning measures fits the human data worse than the model with the four measures (F (4, 5,094) = 7.883, p < 0.0001). Thus, the addition of the four measures improves the fit of the model to data. Our conclusion is therefore that humans are sensitive to some form of head/onset-conditioning: that is, head/onset-conditioning does have psychological reality.

Question 3: is the conditioning orthographic or phonological or both?

What form of head/onset-conditioning is it? It appears at first sight that measure 1 (an orthographic measure) and measure 2a (a phonological measure) are both important, from which one might conclude that the answer to Question 3 is “Both.” However, we are doubtful about such a conclusion because of the high degree of multi-collinearity among the four head/onset-conditioning measures, which we think could be problematical for the reasons given below, and which we sought to circumvent using Principal Components Analysis (PCA) as a means for answering Question 3.

Multi-collinearity is a situation when two or more predictors in a regression model are highly correlated, so that at least one predictor is a linear function of other predictors. In our case, multi-collinearity is present: measures 1–2c are all highly correlated (Table 2). Condition number, which is a measure of multi-collinearity (Baayen, 2008), is 39.5, indicating potentially harmful collinearity. Multi-collinearity is of little concern if one is interested in the overall model fits, but it is problematic if one is interested in the effects of individual predictors, and we are primarily interested in these. In a multiple regression, individual effects of predictors are estimated by holding all other predictors constant (shared variance between predictors is ignored). If other predictors are held constant, and at least one of these is highly correlated with the predictor to be estimated, then less information will be available for the analysis of this specific predictor. This leads to an increased risk of Type II error (a false negative). This is why excluding non-significant predictors from the model that suffers from multi-collinearity may not be a good idea; one is running a risk of losing information that is actually important.

Table 2 Pearson’s correlation coefficients for four head/onset-conditioning measures and type body-consistency.

All correlations are significant at p < 0.0001.

	0	1	2a	2b	
Type_Body					
Measure 1	0.40				
Measure 2a	0.42	0.93			
Measure 2b	0.46	0.85	0.91		
Measure 2c	0.71	0.56	0.60	0.66	

Is multi-collinearity harmful for our analysis? Yes, it is. Using a diagnostic called “tolerance” (Tabachnick & Fidell, 2001), the amount of information that an individual predictor provides for the regression analysis can be estimated. Values that approach zero indicate that there is little such information. For the onset-conditioning measures 1–2c, the tolerance values are 0.13, 0.08, 0.15, 0.35, respectively. This means that, say, for measure 2a, parameter estimates, confidence intervals and significance tests are estimated using only 8% of the available information. The sample size would need to be increased 12.5 times (1/0.08) to overcome the multi-collinearity problem associated with measure 2a. Nonetheless, this 8% of unique information provided by measure 2a influences the overall model; if the measure is omitted, the estimates of other effects change dramatically (for instance, measure 2c becomes significant (B = −0.015, SD = 0.006, t = − 2.7, p < 0.01), which it was not when all four raw variables were considered, see Model B).

Therefore, in the analysis of the raw variables reported above, the effects of the variables are estimated based on little information because of the multi-collinearity problem. One cannot be sure what information is important and what is redundant by looking at the estimates for the model, because these are highly unstable. To meet this problem, PCA was conducted on the four measures of onset-conditioning. PCA is a technique that transforms the original data orthogonally, and yields a set of new, linearly uncorrelated variables (Baayen, 2008; Cohen & Cohen, 1983). PCA removes multi-collinearity from the data. In spite of potential difficulties associated with the interpretation of PCA components, this technique turned out to be useful in our case, because it helped establish which forms of onset-conditioning are having an effect on humans, and which are not.

ELP analyses with principal components

PCA yielded four orthogonal components (components 1–4) that were used instead of the raw values in all subsequent analyses. Note that body-consistency was not included in the PCA analysis, because this measure is not related to head/onset-conditioning. Variance accounted for by each component 1–4 was 82%, 13.2%, 3.5%, 1.3%, respectively.

Multiple regression analyses were performed again on the ELP data, but this time four PCA component measures (Comp_1, Comp_2, Comp_3, and Comp_4) were used instead of original onset-conditioning values (measures 1–2c). Specifically, Model A remained unchanged, Model B’ used the formula lm(I_NMG_Zscore ∼Length + Log_Freq_HAL + Type_Body + Token_Body + Ortho_N + PLD + BG_Freq_By_Pos + Reg + Onset_Length + Type_CV + Token_CV + Affricative + Alveolar + Bilabial + Dental + Fricative + Glottal + Labiodental + Liquid + Nasal + Palatal + Stop + Velar + Voiced + Comp_1 + Comp_2 + Comp_3 + Comp_4), and Model C’ was constructed by excluding non-significant predictors from Model B’.

Model B’ provided a better fit to the data than Model A (F (4, 5,094) = 7.883, p < 0.0001). Among the variables of interest (principal components) in Model B’, only Comp_3 and Comp_4 produced significant effects on human RT. Thus, Model C’ included these two components and did not include Comp_1 and Comp_2 that were not significant in Model B’. Further, we excluded non-significant Token_Body, BG_Freq_By_Pos, CV_type, and Liquid variables. Models B’ and C’ statistically did not differ from each other, so the simpler Model C’ was chosen (p > 0.4). Table 5 presents the output of this final model. We will discuss the effects or non-effects of each principal component below and explain how these advance our understanding of head/onset-conditioning.

Interpretation of principal components and their effects and non-effects

In order to be able to interpret the effects of principal components, the relationship between these components and the original variables had to be established. To do this, the correlations of the principal components with the original measures were computed (Table 3) as were their loadings (Table 4). The correlations show whether the components are directly or inversely proportional to the original variables. The loadings show how much information from the original variables is present in each component. For example, 26% of information in Comp_1 is from Measure 1 (Table 4) and these are inversely correlated: the higher the values of Measure 1, the lower the values of Comp_1 (Table 3).

Table 3 Correlations of principal components with measures for type body-consistency.

	Comp_1	Comp_2	Comp_3	Comp_4	
Type_Body	−0.54	0.46	−0.05	0.01	
Measure 1	−0.94	−0.24	−0.23	−0.11	
Measure 2a	−0.96	−0.21	−0.01	0.18	
Measure 2b	−0.95	−0.07	0.26	−0.08	
Measure 2c	−0.76	0.65	−0.06	0.01	

Table 4 Loadings of the principal components on the original variables.

	Comp_1	Comp_2	Comp_3	Comp_4	
Measure 1	0.26	0.21	0.39	0.29	
Measure 2a	0.27	0.18	0.02	0.48	
Measure 2b	0.26	0.06	0.49	0.21	
Measure 2c	0.21	0.56	0.10	0.02	

Table 5 Output of the final model with principal components as predictors (Model C’).

	B	SE	t value	p value	
(Intercept)	0.087599	0.03453	2.537	p < 0.05	
Length	0.030918	0.003518	8.79	p < 0.001	
Log_Freq_HAL	−0.042211	0.001187	−35.557	p < 0.001	
Type_Body	−0.018652	0.002764	−6.748	p < 0.001	
Ortho_N	−0.026094	0.00353	−7.393	p < 0.001	
PLD	−0.020633	0.003284	−6.283	p < 0.001	
Reg	−0.07278	0.008371	−8.695	p < 0.001	
Bilabial	−0.25672	0.032959	−7.789	p < 0.001	
Dental	−0.404958	0.038478	−10.524	p < 0.001	
Labiodental	−0.404156	0.034641	−11.667	p < 0.001	
Alveolar	−0.196955	0.032159	−6.124	p < 0.001	
Palatal	−0.107347	0.015867	−6.765	p < 0.001	
Velar	−0.214756	0.032673	−6.573	p < 0.001	
Glottal	−0.541422	0.035478	−15.261	p < 0.001	
Nasal	0.118159	0.012166	9.712	p < 0.001	
Stop	0.093491	0.009232	10.126	p < 0.001	
Affricate	0.288089	0.022596	12.75	p < 0.001	
Fricative	0.318311	0.011746	27.099	p < 0.001	
Voiced	−0.040418	0.006823	−5.923	p < 0.001	
Onset_Length	0.009782	0.003364	2.908	p < 0.01	
CV_token	0.00505	0.002563	1.97	p < 0.05	
Comp_3	−0.017602	0.006524	−2.698	p < 0.01	
Comp_4	−0.051565	0.010887	−4.736	p < 0.001	

Component one (Comp_1)

Comp_1 is a linear combination of all four original variables, it also captures the most variance in them (82%). In addition, this is the component that has the highest correlation with type body-consistency. Comp_1 takes the value of 1 for the majority of the items, and these items are also consistent (values for measures 1 and measures 2a–c are equal to 1); and it takes a value different from 1 in inconsistent items (values for measure 1 and measures 2a–c are less than 1). Therefore, Comp_1 distinguishes body-consistent words from body-inconsistent words, i.e., it is a proxy for body-consistency (note that body-consistency was not used for deriving principal components; thus, this variable has emerged by itself from the combination of four onset-conditioned values for body-consistency). Component 1 did not explain variance in human data (note that when type and token body-consistency was excluded from model B’, Component 1 became significant: t = 4.377, p < 0.0001; this indicates that body-consistency and Component 1 indeed contain overlapping information, although body-consistency is a more informative measure than Component 1). Comp_1 duplicates the information contained in the body-consistency measure, therefore its effects or non-effects are not diagnostic with respect to head/onset-conditioning.

Component two (Comp_2)

Comp_2 is unambiguously related to 2c: it takes high values when 2c is high, while other variables are low (in words that become consistent when grouped together by their last onset phoneme’s manner of articulation). An example of words with high Comp_2 values are BASE and CASE; these are OC-consistent when they are grouped together based on the fact that both /b/ and /k/ are stops, but are otherwise body-inconsistent. An effect of Comp_2 would provide evidence for the phonological nature of onset-conditioning (i.e., manner of articulation of the last phoneme in the onset influences sublexical translation), but it is not diagnostic with respect to head-conditioning, because it does not allow for teasing apart of measures 1 and 2a. Component 2 did not explain variance in human reading-aloud data.

Component three (Comp_3)

Comp_3 is mostly related to 1 and 2b: it takes high values when 2b is high and 1 is low, while it takes low values when 2b is low and 1 is high; thus, this component primarily differentiates grapheme OC-consistency (measure 1) from phoneme body-consistency conditioned by place and manner of articulation of the last phoneme in the onset (measure 2b). A facilitatory effect of Comp_3 would be evidence for onset-conditioning (by place and manner of articulation), and against head-conditioning; and vice versa, an inhibitory effect of Comp_3 would provide evidence for head-conditioning, and against onset-conditioning (by place and manner of articulation), because such an effect would suggest that words that are consistent by measure 1 are read faster regardless of other factors (i.e., phonological).

Component 4 (Comp_4)

Comp_4 is related to original measures 1, 2a, and 2b: it takes high values when phoneme OC-consistency is high (2a) and OC-consistency by place and manner of articulation (2b) is low, while it takes low values when phoneme OC-consistency is low and grapheme-conditioned consistency (1) is high. Thus, this component marks out words for which onset-conditioning is more effective at the level of single phonemes (measure 2a) than at the level of graphemes or phonemes arranged into groups by place and manner of articulation (measures 1 and 2b). Since the contribution of measures 1 and 2b is difficult to tease apart, an effect of this component would only support the notion of onset-conditioning, whilst it would not be diagnostic with respect to head-conditioning.

Critically, we found a facilitatory effect of Comp_3 on reading latencies (an inhibitory effect would provide evidence for head-conditioning). Thus, this analysis indicates that there are phonological influences on the form of conditioning we are studying, but provides no evidence of orthographic influences: this is our answer to our Question 3. Therefore, we can at this point abandon the ambiguous term “head/onset-conditioning” and just use the term “onset-conditioning.”

Discussion

Three questions were addressed in the present paper. The first question was: at the level of corpus analysis, does taking the last element in the head/onset reduce the number of possible body-to-rime variants? The answer to this question is positive. In particular, the percentage of body-inconsistent words in the corpus drops from ca. 30% down to 3.5–12% when body-to-rime translation is constrained by the last element in the head/onset.

The second question was: do humans exploit head/onset-conditioning in oral reading? The analysis of the ELP RT showed that a model without the four onset-conditioning measures fits the human data worse than the model with the four measures when other commonly studied psycholinguistic variables like frequency and length are taken into account. Thus, the answer to this question is also positive, i.e., humans are definitely sensitive to some form of head/onset-conditioning when reading aloud.

Next, we attempted to understand which form of conditioning—head or onset—is used by human readers. Multiple regression analysis with the original conditioning measures could not answer this question, because this suffered from a multi-collinearity problem. We therefore used the PCA analysis to circumvent this problem. There are three findings from the regression analysis of the ELP reading-aloud data with principal components as predictors that speak to this question and all indicate that onsets, but not heads, condition the sublexical translation.

First, PCA Component 2 did not explain variance in human data, which suggests that humans do not group phonemes by manner of articulation in oral reading (e.g., stops /b/ and /k/ are perceptually dissimilar). Thus, the similar behaviour of phonemes with common manner of articulation in the corpus (/m/ and /n/ in MOOD, SNOOD vs /g/ and /t/ in GOOD, STOOD) seems to go unnoticed in humans. Even though the grouping by manner articulation could be beneficial for the readers, because phonemes that share the manner of articulation may condition the pronunciation of bodies in a similar way, humans do not use it.

Second, PCA Component 3 explained some variance in human data. Specifically, humans responded faster when they read words with high values of this component. This means that humans RT are faster for words that are consistent when their last phonemes’ place and manner of articulation are considered, compared to words that are inconsistent when their last phonemes’ place and manner of articulation are considered. Consider the example of SCONE. This word has two body-friends, GONE and SHONE. SCONE is read faster than the word DONE which also has two body-friends, NONE and ONE, presumably, because SCONE has an onset-conditioned friend GONE (/k/–/g/ are grouped together by place and manner of articulation), while DONE has no friends onset-conditioned by place and manner of articulation. SCONE is therefore more consistent compared to DONE (from the onset-conditioning perspective). Other examples of onset-conditioning by place and manner of articulation are pairs PEAR/PHASE (PEAR has an onset-conditioned friend BEAR, while PHASE has none), POLL/TOLL (POLL has an onset-conditioned friend BOLL, while TOLL has none) and TEAR/SPADE (TEAR has an onset-conditioned friend DEAR, while SPADE has none).

This finding aligns well with the results of a masked priming study by Mousikou, Roon & Rastle (2015). They found that when initial phonemes in the target and in the prime differ just by one feature (voicing vs devoicing; e.g., /b/ vs /p/, /d/ vs /t/, /f/ vs /v/, /g/ vs /k/, /s/ vs /z/), there is a masked onset priming effect: target reading-aloud latencies are shorter compared to an unrelated control condition (BAV primed with biz is responded to faster than BAV primed with pez, which is in turn faster than in the control condition, i.e., BAV primed with suz). Similarly, our study shows that phonemes that share the place and the manner of articulation condition the rimes, and humans are faster to read words where onset-conditioning is effective than words where onset-conditioning is ineffective. Phoneme pairs in Mousikou, Roon & Rastle (2015) study (/b–p/, /d–t/, /f–v/, /g–k/, /s–z/) are equivalent to five out of seven classes that distinguish the measure 2b from the measure 2a in this study (two additional classes in this study are and ). That is, the grouping by both place and manner of articulation in this study resulted in phoneme pairs that are different on the voicing feature, but similar on all other features. The conclusion is that phoneme pairs that differ in voicing but share place and manner of articulation are perceptually very similar for humans. These units produce the masked onset priming effect (Mousikou, Roon & Rastle, 2015) and also condition the translation of bodies to rimes in oral reading in a similar way.

This finding suggests that phonological similarity of onset-final phonemes (e.g., in terms of phonological features) influences the way sublexical units are stored and retrieved from the lexicon for oral reading. From a rule-based perspective, this means that bodies are translated into rimes using rules that are sensitive to phonological context. From a connectionist perspective, there might be less competition for SCONE and GONE than for DONE, presumably, because within the –ONE body-neighbourhood, there are sub-neighbourhoods arranged by phonological features, and it happens that the body-neighborhood of [+velar, +stop]-ONE is more consistent than the body-neighborhood of [+alveolar, +stop]-ONE. Computational models of oral reading may need to include phonological features as units of representation in order to account for onset-conditioning. This would allow for phonologically similar units like /b/ and /p/ to activate each other, and phonologically less similar units like /b/ and /k/ to have less mutual influence. In theory then, onset-conditioned GPC or body-rime rules in DRC, and feature-based representations in connectionist models could help capture some of the variance in ELP RTs.

The third finding was that humans read words with higher values of Component 4 faster than words with lower values of Component 4. Words with high values of Component 4 have at least one onset-conditioned enemy by place and manner of articulation (e.g., DONE has an enemy STONE, BUT has an enemy PUT, BOOK and an enemy SPOOK, TAIL has an enemy DAIL). On the other end of the continuum are words with low values of Component 4. These are words that are less consistent when their last phonemes in the onset are taken into account (measure 2a) than when their last graphemes in the head are taken into account (measure 1). This means that such words have at least one phoneme-conditioned enemy (WHOSE has en enemy HOSE, SWEAT has an enemy WHEAT, WREATH has an enemy BREATH, WERE has an enemy WHERE).

To illustrate, words like WHEAT are read aloud more slowly than words like STOVE that are matched on regularity and consistency. WHEAT experiences strong competition with SWEAT (WHEAT and SWEAT share the last onset phoneme /w/), while STOVE experiences competition with DOVE (last phonemes in the onset of STOVE and DOVE share place and manner of articulation). Nonetheless, the competition experienced by WHEAT is stronger than that experienced by STOVE. Component 4 exerts an effect independent of the effect of onset conditioning by place and manner of articulation (Component 3 in the analyses; Components 3 and 4 are orthogonal to each other, i.e., they are not correlated, and both explain variance in ELP RT). Difference between WHEAT and STOVE cannot be accounted for using just the two concepts of regularity and consistency, because WHEAT and STOVE do not differ in regularity or in consistency. Other examples of this kind are HOSE > PAST (HOSE experiences a strong competition from WHOSE, both share the last onset phoneme /h/; while PAST experiences a weaker competition from BAST); WHERE > TOUGH (WHERE experiences a strong competition from WERE, both share the last onset phoneme /w/; while TOUGH experiences a weaker competition with DOUGH); BREATH > PHASE (BREATH experiences a strong competition from WREATH, both share the last onset phoneme /r/; while PHASE experiences a weaker competition from VASE). One potential caveat here is that most onsets in words with low values for Component 4 are complex onsets (indeed, grapheme- and phoneme-based consistencies would diverge in words that contain multi-letter graphemes, like WH vs /w/, KN vs /n/).

The main problem with these analyses is that components of interests (3 and 4) that came out as significant in subsequent analyses capture little variance in the raw data: together they account for around 5% of variance. Therefore, one could argue that onset-conditioning may not be very important for oral reading overall and can be safely ignored. While it is true that Components 3 and 4 only influence about 13% of the analysed corpus (i.e., 680 words have a value less than 1), our primary interest is not in the corpus, but in human behaviour. Obviously, components 3 and 4 are not related to body-consistency (see Table 3). These components are uniquely related to head/onset-conditioning and include no other information. Further, the effect sizes for Components 3 and 4 are small (0.12% and 0.44% of variance explained; although comparable to other commonly studied variables like Type_body 1.61%, Reg 0.55%, Onset_Length 0.22%). Nonetheless, the components explain a significant amount of variance in the human data. Therefore, we feel confident in concluding that in human readers, the translation of bodies to rimes is influenced by the last element in the onset.

The effects of principal components should be interpreted with caution, because these comprised several different measures whose influence cannot be teased apart. Nonetheless, in our case PCA was useful, because it advanced our understanding of head/onset-conditioning in three major ways. First, PCA suggests that human readers do not exploit onset-conditioning by manner of articulation (Component 2; e.g., they do not benefit from the fact that both /b/ and /k/ are stops in BASE and CASE). Second, the conditioning of body pronunciation seems to take place in phonology rather than orthography. Had humans exploited head-conditioning, one would observe an inhibitory effect of Component 3. If humans relied on graphemes rather than phonemes, they would be faster reading words like PHASE compared to words like PEAR (cf. SPEAR) regardless of other factors. Nonetheless, humans do rely on phonemes and/or phonological features, benefitting from the similarity between BEAR and PEAR, and cannot avoid the competition between PHASE and VASE. Third, PCA confirmed that body-to-rime translation in oral reading is conditioned by the last phoneme in the onset and/or its phonological features such as place and manner of articulation (effects of Components 3 and 4). Onset-conditioning by this factor is used automatically by human readers: this leads to faster responding to words like SCONE and GONE and to slower responding to words like STONE and DONE.

Our study does not speak to questions about the nature of reading units. First, we have explored onset-conditioning at the level of bodies and rimes. But this does not mean that onset-conditioning occurs at the level of bodies and rimes. It is equally possible that the last element in the onset conditions the pronunciation of the following vowel, not the whole body, so that onset-conditioning may be the property of phoneme-to-phoneme dependence. In other words, we used the phonological properties of /p/ to predict the pronunciation of –USH, but we could instead have used the phonological properties of /p/ to predict the pronunciation of –U–. Calculating both and disentangling their effects on oral reading will help uncover the underlying nature of onset-conditioning. Second, as explained in the ‘Introduction,’ antibody- and head-consistency are phenomena that are different from onset-conditioning. Unlike with antibody- or with head-consistency, the effect of onset-conditioning should not be taken as evidence for the formation of larger grain-size reading units, such as the antibody or the head. To illustrate, the finding that /p/ conditions –USH, does not mean that humans rely on large units like PU–. Our study investigates the dependency between word-beginnings and word-ends, and does establish the psychological reality and phonological nature of such dependency in English oral reading.

Our study also does not speak to the question of which of the various stages of the reading-aloud process is the one at which the onset-conditioning effect could be arising. In particular, our findings provide no information as to whether the onset-conditioning effect arises during the process of activating phonology from print, or during the process of creating an articulatory plan from a phonological representation.

Conclusions

We have shown that there is a dependency between word-beginnings and word-ends in English (onset-/head-conditioning) at the level of corpus analysis. In other words, the pronunciation of English bodies is dependent on the preceding element. We demonstrated the psychological reality of this phenomenon by performing an analysis of reading-aloud latencies times from the English Lexicon Project database. In particular, we found that phonological, rather than orthographic units condition the pronunciation of bodies in human oral reading.

Supplemental Information

Supplemental Information 1 Raw Data

Click here for additional data file.

We thank Steven Saunders for his help with programming.

Additional Information and Declarations

Competing Interests

Author Contributions

Data Availability

1 A grapheme is a letter or letter-sequence that corresponds to a single phoneme (e.g., F, PH, IGH, EIGH). Graphemes were defined based on a non-rhotic dialect of English.

2 A body is a letter sequence in a monosyllabic letter string from its vowel to its end. It corresponds to rime in phonology. A head is a letter sequence in a monosyllabic letter string from the beginning up to its vowel (not including the vowel). It corresponds to onset in phonology.

3 The antibody of a monosyllabic letter-string is its letters up to and including its vowel letter(s); for example, the antibody of SHEATH is SHEA (and the antibody of WEIGH is WEIGH).

4 A body-enemy is a word whose body is pronounced differently than in the target word. A body-friend is a word whose body is pronounced in the same way as in the target word.

The authors declare there are no competing interests.

Anastasia Ulicheva conceived and designed the experiments, analyzed the data, wrote the paper, prepared figures and/or tables, reviewed drafts of the paper.

Max Coltheart conceived and designed the experiments, wrote the paper, reviewed drafts of the paper.

The following information was supplied regarding data availability:

The research in this article did not generate any raw data.

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
