# Peer review of "How word-beginnings constrain the pronunciations of word-ends in the reading aloud of English: the phenomena of head- and onset-conditioning"

_PeerJ, doi:10.7717/peerj.1482_

## Round 0.1 · original submission · Major Revisions

I have received two expert reviews on your manuscript, and there is a high degree of agreement between the two. Both note that your findings were not adequately grounded in the extant literature on word perception (in general) and consistency (in particular). Reviewer 1 has a series of analytical concerns that need to be addressed in a revision, and Reviewer 2 (Greg Stone) suggests more clarity in the presentation of statistical analyses and reporting of outcomes.

While Reviewer 1 expressed some concern about the inclusion of the linear models, I feel that the analyses contribute important procedural details to the narrative (and do not take up too much real-estate within the manuscript). Consequently, I do not believe that they need to be removed.

I look forward to receiving a revised version of this work.

Reviewer 1 ·

Basic reporting

This project needs to be better situated in existing research. There are a number of studies that look at predictors of naming latencies from large data bases, and these have included C1V1 consistency (e.g., Treiman, Mullenix, Bijeljac-Babic, & Richmond-Welty, 1995; Balota, Cortese, Sergent-Marshall, & Spieler, 2004; Chateau & Jared, 2003; Yap & Balota, 2009). The reader of this paper will likely know about those studies and wonder how this work extends these previous studies. In those studies C1 was taken to be all the consonants before the first vowel. If I am understanding this work correctly, in the case of initial letter clusters, only the final grapheme/phoneme was considered. Why is not considering the other letters in the cluster (if any) likely to be more informative regarding the pronunciation of following letters?

Experimental design

(the second criterion here is addressed with comments on the third criterion above)

Validity of the findings

The presence of W has a predictable impact on the pronunciation of a following A. If words with a W as the last letter before an A are removed from the list of items on which the analyses were conducted, is the same pattern of results still evident? That is, to what extent are the findings due to WA words? To what extent are results influenced by predictable G and C pronunciation changes in the presence of E and I?

A token measure of consistency was used. This should NOT be referred to as a “traditional” body-consistency measure, as some other studies use type based measures. To avoid confusion, call it what it is- token body consistency. Are similar results obtained when a type based measure is used?

The analyses should include a variable coding the number of letters in the onset. This variable was a significant predictor in the Treiman et al study.

I’m not sure readers are going to be happy about wading through the first set of analyses only to be told that multi-collinearity was an issue, and a new set of analyses done. Why not just the second set?

Additional comments

Why is Table 4 first?

When Table 1 is referred to, there is the comment that condition number is 39.5- what does that mean? Table 1 doesn’t have a number 39.5.

Discussion, second page, line 45. Should the < be the other way (i.e. > )? I would guess that priming effects for biz-BAV would be larger than for pez-BAV

Introduction, first sentence. A reference is needed to the DRC. Words are only regular or irregular with respect to a specific set of rules.

·

Basic reporting

The authors present an intriguing analysis of the ELP corpus. The analysis deserves to be published, but I would like to see the authors address 3 issues first.
First, they are very light on their review of the word recognition field. There is quite a history to the regularity versus consistency distinction, which should be discussed. There is also a relevant literature on analysis of the role of various subword units in spelling-sound relationships (the Trieman et al. paper on the special role of rimes, for example). In particular, I recall that Seidenberg did work looking at “anti-body” effects, which is very similar to this analysis. It’s relationship to this analysis should be discussed. Explain briefly what ELP is. I can’t recall right now if ELP adjusted RTs for initial phoneme enunciation effects in naming. If so, mention that, if not, include the appropriate initial phoneme variables in the analysis. This is particularly important as initial phoneme of the word and final phoneme of the onset are usually one and the same.
On a related point, the conclusion is more a rather technical summation than a proper concluding discussion. I skipped ahead to it for the “take home message” but didn’t really get one.
Second, although the authors dealt with the issue of multicollinearity in an appropriate fashion (under the circumstances, PC analysis was the best choice), the presentation was extremely hard to follow. I had to dust off my Cohen & Cohen to get through and remind myself why they were doing the analyses they were doing and then to understand what was going on. The reader will really need to feel comfortable with their understanding of the findings, especially my final issue.
Third, the authors should report effect sizes. With a df greater than 5000, p < .001, is not in itself interesting. If the effect sizes are moderate to large (which I doubt), the multicollinearity problem is less of a concern. If they are small, they may still be interesting, but the multillinearity problem becomes a much more important caveat.

Experimental design

Not applicable? Not an experiment, but the data are publicly available and The R code would allow anyone to check their analysis

Validity of the findings

We need effect sizes to judge robustness. If the authors want to pursue this point then, in a separate publication, they should "replicate" in a carefully designed experiment.

---

## Round 0.2 · Minor Revisions

I have, once again, received two expert reviews on your manuscript. While both reviewers noted improvements in the manuscript, they also indicated that the arguments you have presented lack clarity and that the theoretical grounding of your paper is still lacking.

In particular, I'd like to draw your attention to the comments from Reviewer 1. Like Reviewer 1, I found your new paragraph on page 4 to be intractable, and had a difficult time aligning your results with the greater word perception literature. While PeerJ's criteria for acceptance are biased toward methodological rigor, it is exceedingly difficult to appreciate your results without a more coherent theoretical framework. I hope you will try to enhance the clarity of your manuscript and submit a revised document.

Reviewer 1 ·

Basic reporting

Criterion:
The article should include sufficient introduction and background to demonstrate how the work fits into the broader field of knowledge. Relevant prior literature should be appropriately referenced.

In my last review I said that this project needs to be better situated in existing research. The references that I had suggested have been added but with very little explanation.

Experimental design

Criterion:
The submission should clearly define the research question, which must be relevant and meaningful. The knowledge gap being investigated should be identified, and statements should be made as to how the study contributes to filling that gap.

I still find the rationale of this paper to be quite weak.

The points made in the paragraph added to p. 4 are not very clear. I don’t understand the explanation of the difference between head/onset conditioning and CV consistency. Don’t they both refer to the extent to which initial letters constrain the pronunciation of the following vowel? And I’m confused by the CLOW segment of CLOWN and the HEAR segment for HEARD and HEART in the examples. I suppose the W is part of the vowel in the first case (not the best choice of examples), but why is the R included in the second case? The CV is HEA. HEAR, HEARD and HEART have different word bodies. The explanation here needs to be clearer.

It is indicated in the introduction that 66% of words have simple onsets. This work looks at whether the word body is more consistent if aspects of the consonant before the vowel are taken into account. But why wouldn’t it be? For most words, isn’t this considering the whole word?

I think that there needs to be clearer exposition of the theory on which this work is based. If I’m understanding correctly, for measures 2a-c the consonant preceding the vowel needs to be converted into its sound form first before it can provide help in determining the pronunciation of the vowel. This assumption of serial assignment of phonemes to graphemes needs to be made more explicit and evidence provided to support the claim. The DRC, of course, makes this assumption. But what is the mechanism in that model that would allow knowledge of a phoneme to help in selecting the appropriate rule for a following grapheme? And if processing is serial, why or how are the consonants at the end of the word body considered?

A relevant study by Mouikou et al. (2015) only appears in the discussion. Could it not help build the rationale for this study?

There needs to be some discussion of whether measures 2a-c influence the computation of phonological representations or are related to articulatory output.

In summary, the reader is going to want to see a much clearer and more convincing rationale for the study before they will wade into the remainder of the paper.

Validity of the findings

No comments

Additional comments

No other comments.

·

Basic reporting

The review of relevant literature is much better now.
In the last manuscript, it took too much effort to really understand the difference between onset conditioning and CV consistency. The 1st paragraph on page 4 really helped with this.
The reporting is clear and unambiguous, but still requires a lot of effort to follow. This isn't an impediment to publishing, but may reduce readership.

Experimental design

The data are in a public database. The question was relevant to the field and the analyses were conducted at a high standard.
Possible confounds with initial phoneme effects was a serious concern, but they have been addressed in the new analysis.

Validity of the findings

The use of PCA reflected a higher standard than is usually found in large linear model analyses in the field, and the interpretation of principle components was appropriately circumspect.

Additional comments

If the effect holds up, it is of interest to the field, but I worry your audience won't find this paper. The title gives no sense of what the paper is about. I would recommend you choose a title that refers to onset conditioning, or makes a more direct reference to the concept.
I still recommend following up with an experimental demonstration of the effect, as this would be much more compelling.

---

## Round 0.3 · accepted · Accept

I appreciate the work that you have put into this manuscript. I believe that it has improved substantially across revisions. I applaud your for you diligence. This is a valuable contribution to the literature.